# Fulvic Acid Improves Salinity Tolerance of Rice Seedlings: Evidence from Phenotypic Performance, Relevant Phenolic Acids, and Momilactones

**DOI:** 10.3390/plants12122359

**Published:** 2023-06-18

**Authors:** Akter Jesmin, La Hoang Anh, Nguyen Phuong Mai, Tran Dang Khanh, Tran Dang Xuan

**Affiliations:** 1Graduate School of Advanced Science and Engineering, Hiroshima University, 1-5-1 Kagamiyama, Higashi-Hiroshima 739-8529, Japan; js_jesmin@yahoo.com (A.J.); hoanganh6920@gmail.com (L.H.A.); maiphuongnguyen.1094@gmail.com (N.P.M.); 2Department of Agricultural Extension, Ministry of Agriculture, Dhaka 1215, Bangladesh; 3Center for the Planetary Health and Innovation Science (PHIS), The IDEC Institute, Hiroshima University, 1-5-1 Kagamiyama, Higashi-Hiroshima 739-8529, Japan; 4Agricultural Genetics Institute, Pham Van Dong Street, Hanoi 122000, Vietnam; 5Center for Agricultural Innovation, Vietnam National University of Agriculture, Hanoi 131000, Vietnam

**Keywords:** fulvic acid, salinity, rice, antioxidants, chlorophylls, phenolics, momilactones

## Abstract

Salinity is a severe stress that causes serious losses in rice production worldwide. This study, for the first time, investigated the effects of fulvic acid (FA) with various concentrations of 0.125, 0.25, 0.5, and 1.0 mL/L on the ability of three rice varieties, Koshihikari, Nipponbare, and Akitakomachi, to cope with a 10 dS/m salinity level. The results show that the T3 treatment (0.25 mL/L FA) is the most effective in stimulating the salinity tolerance of all three varieties by enhancing their growth performance. T3 also promotes phenolic accumulation in all three varieties. In particular, salicylic acid, a well-known salt-stress-resistant substance, is found to increase during salinity stress in Nipponbare and Akitakomachi treated with T3 by 88% and 60%, respectively, compared to crops receiving salinity treatment alone. Noticeably, the levels of momilactones A (MA) and B (MB) fall in salt-affected rice. However, their levels markedly rise in rice treated with T3 (by 50.49% and 32.20%, respectively, in Nipponbare, and by 67.76% and 47.27%, respectively, in Akitakomachi), compared to crops receiving salinity treatment alone. This implies that momilactone levels are proportional to rice tolerance against salinity. Our findings suggest that FA (0.25 mL/L) can effectively improve the salinity tolerance of rice seedlings even in the presence of a strong salt stress of 10 dS/m. Further studies on FA application in salt-affected rice fields should be conducted to confirm its practical implications.

## 1. Introduction

Rice (*Oryza sativa*) is the third most widely cultivated crop globally, with a production of 519.5 million tons in 2022 [1]. Rice serves as a staple food for the majority of the world’s population, meeting up to 35–75% of their daily calorie requirements [2]. Salinity is a serious environmental stress that affects 1.26 billion hectares of the world’s land and 1.5 million hectares of agricultural land annually, leading to reduced production [3]. This is a major threat to crops, especially in the case of rice, causing a 70% reduction in yield [4]. In order to cope, rice has the natural ability to adapt to salt stress through mechanisms such as ion exclusion, osmotic stress tolerance, and tissue tolerance [5]. Advances in rice breeding and genetic engineering have enabled the identification of specific salt-tolerance genes and the development of new tolerant rice varieties through the integration of QTLs, genetic modifications, and mutations. While these efforts have shown success, the development and production processes are complex and time-consuming. Furthermore, as salt tolerance is regulated by multiple genes and is a quantitative trait, undesirable mutations may occur that require additional correction efforts [6]. Additionally, environmental factors may cause gene silencing in transgenic plants [7,8]. Despite offering resistance to environmental stress, the quality and yield of most salt-tolerant varieties do not meet expectations [9,10]. Therefore, finding new cultivation methods and supplementary products to help rice better cope with salt stress is a top priority. In addition, as natural disasters are often unpredictable, on-site treatments are considered more effective. For instance, the external application of fertilizers and/or bio-preparations may help rice recover from salinity stress. Researchers have explored the use of natural secondary metabolites to improve rice’s tolerance to salinity [11,12], however, their physical properties, such as solubility in water and pH stability, limit their effectiveness. Thus, further research is necessary to find new and efficient substances for the topical treatment of rice under salt stress. In this context, natural products are a favorable choice as they are non-toxic and highly effective.

Fulvic acid (FA) is a naturally occurring compound and an example of a humic substance, which persists in soil for extended periods and aids in the growth of plant roots [13]. FA has also been reported to promote primary and secondary metabolites in numerous plants, thereby improving their defense against various abiotic stresses, including salinity [14,15]. In recent years, the potential of FA for enhancing crop productivity by alleviating salt stress, improving soil physical characteristics, and nutrient absorption has gained widespread recognition [14]. In particular, FA treatments have stimulated salinity tolerance of crops including soybean [16], almond rootstock [17], yarrow [18], and spinach [19]. Though FA has been recognized for its high potential for enhancing the tolerance of numerous crops to abiotic stresses, its role in improving rice resistance to salinity has not been scrutinized. Therefore, the objective of this study is to determine the effects of FA applications on phenotypic performances, antioxidant activity, and chemical profiles of three rice varieties, Koshihikari, Nipponbare, and Akitakomachi, during salinity stress.

## 2. Results

### 2.1. Phenotypic Performances

#### 2.1.1. Injury Scores and Survival Rates

Salinity exhibits a significant negative impact on the plant growth performance of the Koshihikari, Nipponbare, and Akitakomachi varieties, leading to lower survival rates, higher seedling injuries, and decreasing plant growth performance compared to the control (Figure 1 and Figure 2). The survival rates of these varieties are only 36.67%, 26.67%, and 30%, respectively, in the presence of salt stress, while their injury scores are 4.87, 6.2, and 5.73, respectively. Noticeably, during salinity stress, the application of fulvic acid (FA) at 0.25 mL/L (T3 treatment) is the most effective treatment for promoting the survivability of the Koshihikari, Nipponbare, and Akitakomachi varieties by 1.82, 2.69, and 2.28 times, respectively, compared to crops receiving salinity treatment alone (Figure 2). In addition, the injury scores of Koshihikari, Nipponbare, and Akitakomach treated with T3 are markedly reduced by 1.83-, 4.22-, and 2.97-fold, respectively, over crops receiving salinity treatment alone (Figure 2).

#### 2.1.2. Shoot Height, Root Length, Fresh Weight, and Dry Weight

Remarkably, the application of fulvic acid (FA) enhances plant phenotypic parameters, especially the root growth parameter, of all three rice varieties compared to the control. Once again, T3 (0.25 mL/L FA) gives the best performance among the different treatments. The result shows that, with T3 treatment, Koshihikari seedlings recover their shoot height, root length, fresh weight, and dry weight by 80%, 125%, 70%, and 156%, respectively, compared to the control, while for Nipponbare seedlings the equivalent improvements are 104%, 105%, 78%, and 171%, respectively. Similarly, the enhancement in shoot height, root length, fresh weight, and dry weight of Akitakomachi seedlings are recorded at 92%, 149%, 99%, and 138%, respectively (Table 1). T2 treatment has a minor effect on the recovery of rice seedlings from salinity stress. Differences in shoot height and root length among varieties and treatments are significant, whereas those in fresh weight and dry weight are not significant (Table 1).

### 2.2. Chemical Performances

#### 2.2.1. Chlorophyll Contents

Figure 3 presents the effects of fulvic acid (FA) on the chlorophyll contents of Koshihikari, Nipponbare, and Akitakomachi rice seedlings at 10 dS/m salinity. Generally, salinity lowers the accumulation of chlorophylls a and b and total chlorophylls in all tested rice cultivars. The application of FA has a varying effect on the levels of chlorophylls depending on the cultivar and the specific FA treatment used. For chlorophyll a, FA application has insignificant effects on both Nipponbare and Akitakomachi compared to the control. In contrast, all FA treatments, except T2, stimulate the quantity of chlorophyll a in Koshihikari during salinity stress. The highest amounts of chlorophyll a are recorded under T3 treatment at 31.78, 29.06, and 27.00 μg/g in Koshihikari, Nipponbare, and Akitakomachi, respectively. The results for chlorophyll b are shown in Figure 3B. As can be seen, T2 treatment has no effect on the chlorophyll b contents of all tested cultivars, but the T3, T4, and T5 treatments result in a remarkable increase in chlorophyll b levels in all three varieties during salinity stress. However, there is no significant difference in the effect of these treatments on chlorophyll b levels in all tested rice cultivars under salt stress. Under T3 treatment, chlorophyll b levels are highest in Akitakomachi (48.04 μg/g), followed by Nipponbare (40.87 μg/g) and Koshihikari (32.63 μg/g). Figure 3C presents the total chlorophyll content, which follows the same trend as that of chlorophyll b. Noticeably, all treatments, except T2, exhibit stimulatory effects on the total chlorophylls of all rice cultivars compared to the crops receiving salinity treatment alone. However, the increasing FA concentrations in the T3, T4, and T5 treatments show negligible differences in their effects on total chlorophylls during salinity stress. Under T3 treatment, the total chlorophylls of Koshihikari, Nipponbare, and Akitakomachi are 64.38, 69.90, and 74.59 μg/g, respectively.

#### 2.2.2. Total Phenolic (TPC) and Flavonoid (TFC) Contents and Antioxidant Activity

Salinity significantly decreases total phenolics (TPC) by 56%, 63%, and 49%, and total flavonoids (TFC) by 60%, 47%, and 59% in Koshihikari, Nipponbare, and Akitakomachi rice samples, respectively, compared to the control (Figure 4A,B). In contrast, the highest TPC (1.50 and 1.07 mg GAE/g DW) and TFC (0.33 and 0.48 mg RE/g DW) are found in Nipponbare and Akitakomachi samples receiving T3 treatment (Figure 4A,B). Similarly, salinity treatment reduces the antioxidant activity of Koshihikari, Nipponbare, and Akitakomachi rice seedlings by 53%, 76%, and 68%, respectively, over the control (Figure 4C). Among FA applications, all varieties treated with T2 (0.125 mL/L FA) reveal negligible antioxidant activity under salt stress. Meanwhile, Koshihikari, Nipponbare, and Akitakomachi treated with T3 (0.25 mL/L FA) exhibit the strongest antioxidant activity (IC_50_ = 2.10, 1.91, and 1.48 mg/mL, respectively) during salt stress, levels that are close to those of control seedlings grown in normal conditions (Figure 4C). Significantly, T3 treatment stimulated TPC (2.27- and 1.60-fold), TFC (1.65- and 3.00-fold), and antioxidant activity (2.36- and 1.78-fold) in Nipponbare and Akitakomachi, respectively, compared to crops receiving salinity treatment at 10 dS/m alone (T1).

#### 2.2.3. Phenolic Profiles

The highest TPC, TFC, and strongest antioxidant activity are recorded in T3 samples, which were selected for further chemical analyses to detect their phenolic and momilactone contents. As can be seen in Table 2, all detected phenolic compounds decrease dramatically in salt-exposed seedlings. However, the application of the T3 treatment leads to a recovery in all phenolics except vanillin. Remarkably, Nipponbare seedlings show the greatest recovery in phenolic contents with a remarkable amount of salicylic acid (0.25 mg/g DW), while Koshihikari seedlings exhibit the highest contents of vanillin (0.17 mg/g DW) and cinnamic acid (0.16 mg/g DW). Additionally, caffeic acid is detected in only the T3-treated samples of Nipponbare and Akitakomachi seedlings (Table 2).

#### 2.2.4. Momilactone Contents

Table 3 shows the effects of FA on the contents of momilactones A (MA) and B (MB) in rice samples. T3-treated samples reveal significantly higher MA (3.09 and 5.49 µg/g) and MB (2.64 and 3.66 µg/g) contents in Nipponbare and Akitakomachi, respectively, compared to T1-treated samples. Furthermore, MA increases approximately 2.00- and 3.10-fold while MB increases 1.47- and 1.89-fold in Nipponbare and Akitakomachi, respectively, compared to seedlings receiving salinity treatment alone. However, the momilactone contents decrease (by over 50%) in Koshihikari receiving T3 treatment. Nevertheless, the highest contents of MA (15.47, 12.95, and 7.15 µg/g) and MB (7.42, 12.31, and 7.20 µg/g) are exhibited in the control seedlings of Koshihikari, Nipponbare, and Akitakomachi, respectively. This finding indicates that MA and MB might play a role in improving the salinity tolerance of rice seedlings at 10 dS/m.

## 3. Discussion

Recently, exogenous applications of fertilizers, biostimulants, and bio-preparations for improving plant tolerance to environmental stresses have received increasing attention from researchers worldwide [11,12,20,21]. However, their effectiveness is limited due to physical properties such as solubility and pH persistence [11,12]. Notably, fulvic acid (FA), a small molecule and pH-soluble compound, was shown to improve the tolerance of numerous plant species to stresses, for example, tea and maize against drought conditions [14,15], and spinach plants against salt stress [19]. However, to date, no study has explored the use of FA in rice production to mitigate salinity stress. In the current study, via the investigation of three rice (*Oryza sativa* L.) varieties, Koshihikari, Nipponbare, and Akitakomachi, we demonstrate, for the first time, that FA may improve rice tolerance to salinity conditions by increasing the antioxidant activity, phenotypic performance, and accumulation of reported salt stress-resistant phytocompounds. Furthermore, the effectiveness of bio-preparations is dependent on the applied dosage [12]. Therefore, different FA concentrations (0.125, 0.25, 0.5, and 1.0 mL/L) were examined in our study to determine the most effective dose for helping rice to cope with salinity.

Under salt stress, the phenotypic parameters (shoot height, root length, fresh weight, and dry weight) of rice seedlings significantly declined, which is consistent with previous studies [22,23,24]. However, these phenotypes are remarkably improved under FA treatment. Specifically, FA application may increase the root growth of rice seedlings, which may allow the plant to cope with salinity by accumulating more water and storing it for a longer duration [23]. Notably, T3 treatment (0.25 mL/L FA) is the most effective concentration for increasing rice growth under salt stress. In addition, photosynthesis is an indispensable process in rice. In our study, photosynthetic parameters including total chlorophylls, chlorophylls a and b rapidly deteriorate in rice seedlings subjected to 10 dS/m salinity. In preceding studies, FA was indicated to enhance the contents of the photosynthetic pigments of tomato (*Lycopersicon esculentum* L.) in greenhouse conditions [25] and yarrow (*Achillea millefolium* L.) under both greenhouse and field conditions [18]. In the present research, FA treatment, especially T3, may elevate chlorophyll contents, allowing these to surpass or equal their levels in seedlings grown under normal conditions or receiving salt treatment alone. The finding suggests that FA may help protect chlorophyll levels in rice seedlings, thus improving their photosynthetic ability under salinity stress. Moreover, photosynthesis plays a vital role in regulating transpiration [26], which may increase rice growth during salt stress.

In another aspect, decreased rice growth may be associated with induced oxidative stress through an increase in reactive oxygen species (ROS) due to salt stress [27]. Therefore, antioxidant activity is an integral factor determining rice resistance against salinity. In the current study, although antioxidant activity fell in rice seedlings affected by salinity, it recovered due to the applications of FA, especially T3 treatment. Based on the strengthened antioxidant defense system, rice may fight against salinity by nullifying the detrimental ROS [28]. It should be noted that the antioxidant activity of rice cultivars against stresses may be determined by the activities of relevant enzymes such as superoxide dismutase, catalase, glutathione peroxidase, and ascorbate peroxidase [29]. Therefore, the effects of FA on these enzymes need comprehensive elaboration in future studies. 

Additionally, the rice antioxidant system can be activated due to the proliferation of phenolic compounds, which detoxify free radicals by donating hydrogen ions [30]. The upregulated accumulation of these compounds is considered a mechanism by which numerous plant species, including rice, are able to tolerate abiotic stresses [22,31,32,33]. In the present study, total phenolic (TPC) and flavonoid (TFC) contents are dramatically decreased in salt-affected rice seedlings. However, TPC and TFC levels are higher in FA-treated seedlings. The results imply that FA may elevate the accumulation of rice TPC and TFC, thereby mitigating the injuries induced by oxidative stress during salt stress. Notably, the greatest increases in TPC and TFC were recorded in rice seedlings treated with 0.25 mL/L FA (T3 treatment) under salt stress. Among the phenolic compounds detected in the T3 sample by HPLC analysis, salicylic acid (SA) is found with increased levels compared to the crop receiving salinity treatment alone. It is worth noting that SA has exhibited a potential for increasing plant tolerance to abiotic stresses, i.e., salinity [34,35]. In contrast, the levels of the known stress-resistant phenolics gallic, protocatechuic, cinnamic, and caffeic acids [22,29,31,36] are found in this study to increase only in the strongest salt-tolerant rice variety, Nipponbare, when treated with FA. This might be due to differences in the genetic diversity of the selected rice cultivars. In fact, different rice varieties (tolerant and susceptible cultivars) exhibit dissimilar mechanisms for phenolic accumulation to cope with stress conditions [22,31,36,37], which may result in various responses to FA application. 

In addition to variations in the levels of phenolic compounds, variations in the levels of momilactones A (MA) and B (MB) in rice seedlings treated with FA during salinity stress are indicated in the present research. These compounds have been previously acknowledged to have a high potential for treating various human diseases [38,39,40,41]. In addition, a previous investigation of 30 different rice cultivars indicated that MA and MB may also play a role in enabling rice to withstand salt stress [42]. Notably, the levels of MB appeared to have a stronger association with the plant’s ability to tolerate salinity than the levels of MA [42]. In this study, the levels of MA and MB decreased in salt-affected rice seedlings, which is consistent with the reported results of Xuan et al. [21]. Significantly, our research suggests that FA may increase the levels of MA and MB in Nipponbare and Akitakomachi but decrease those in Koshihikari seedlings under salt stress. However, the role of momilactones in the response of rice varieties to salt stress remains unclear and needs further clarification. In particular, exploring the role of momilactones and their relationship with resistance-associated molecules against salinity, including phytohormones and functional enzymes, could be a promising approach that should be comprehensively scrutinized in future studies. Moreover, to further comprehend the effects of FA treatment on phenolic and momilactone biosyntheses in salt-affected rice, future studies on the expressions of relevant genes should be conducted. The biosyntheses of phenolics and momilactones are particularly closely associated with the regulation of genes encoding the major biosynthetic enzymes [43,44]. Studies have found that the upregulated expression of certain genes (e.g., phenylalanine ammonia-lyase and chalcone synthase) results in an increased production of phenolic compounds, which in turn enhances the ability of rice to withstand abiotic stress, i.e., salinity [33,37]. At the same time, the elevated expression of genes related to momilactone biosynthesis, including OsCPS4, OsKSL4, CYP99A3, OsMAS, and OsMAS2, is strongly correlated with the proliferation of MA and MB, which may contribute to rice tolerance to adverse conditions [37].

In summary, our findings imply that FA at a concentration of 0.25 mL/L (T3) is the most effective treatment for improving rice tolerance against salt stress at 10 dS/m by mitigating plant injuries, increasing survivability rates, and protecting phenotypic performances (shoot and root length, fresh and dry weights). Additionally, T3 treatment significantly elevates chlorophyll levels, thereby enhancing the photosynthetic capacities of all the tested rice varieties under salinity stress. In addition, in our study, T3 treatment also markedly increased the levels of salt-resistant phenolic compounds, especially gallic, protocatechuic, salicylic, cinnamic, and caffeic acids [23,30,31,32,35,36,37], which may mitigate ROS damage by elevating antioxidant activity in salt-stressed rice. Similarly, T3 treatment promotes the proliferation of MA and MB, which is in line with the enhanced tolerance of rice to salinity. The finding suggests a potential role of momilactones in the rice defense system against salt stress that needs further clarification. Our research has implications for protecting rice production in salinity-affected areas. Moreover, the present research is anticipated to support the achievement of the UN Sustainable Development Goals (SDGs) by promoting poverty eradication and food security worldwide, particularly in countries reliant on rice cultivation.

## 4. Materials and Methods

### 4.1. Materials 

Fulvic acid (FA) was donated by Kume Sangyo Co., Ltd., Higashi Hiroshima City, Japan. Extraction solvents were purchased from Junsei Chemical Co., Ltd., Tokyo, Japan. The reagents and chemicals, including sodium hypochlorite (NaOCl), Folin–Ciocalteu’s reagent, gallic acid, rutin, sodium carbonate (Na_2_CO_3_), sodium acetate (CH_3_COONa), aluminum chloride (AlCl_3_), 2,2-diphenyl-1-picrylhydrazyl (DPPH), sodium chloride (NaCl), and hydrochloric acid (HCl), were obtained from Kanto Chemical Co. Inc., Tokyo, Japan. Samples of momilactones A (MA) and B (MB) previously isolated and purified by the Laboratory of Plant Physiology and Biochemistry, Hiroshima University, Japan, were used as standards [38].

Rice seeds of the Koshihikari, Nipponbare, and Akitakomachi cultivars were purchased from Japan Agricultural (JA) Cooperatives, Hiroshima, Japan. Initially, seeds were sterilized by soaking in 0.1% NaOCl for 30 min followed by washing with distilled water several times. Afterward, seeds were immersed in water at 40 °C for 3 days for germination. The germinated seeds were then transferred into a floating tray (100 cm × 200 cm × 80 cm) with two seeds per hole. The tray was filled with 2 L of Yoshida solution and kept in a growth chamber with 30 °C/28 °C day/night and a 12 h photoperiod for one week. Next, NaCl was added to generate an electrical conductivity (EC) level of 10 dS/m, which is considered a high-stress condition and important for studying the response of rice to salt stress [45]. Four different concentrations of FA (0.125, 0.25, 0.5, and 1.0 mL/L) were used as treatments, while trays without NaCl and FA served as control. The FA treatments were conducted in the presence of salinity for 21 days. The EC (10 dS/m) of the growth solution was maintained daily using an EC meter (Hann HI 4321, St. Louis, MO, USA). Meanwhile, the pH (5.5) of the solution was monitored using a pH meter (Thermo Scientific Orion Star A111, Waltham, MA, USA) calibrated in pH measurement mode within the pH range of 4.01 to 7.00. The volume of the growth solution was replenished to 2 L every 2 days. The treatments are described in Table 4. Each treatment was repeated three times.

### 4.2. Phenotypic Evaluation

After 21 days of treatment, 10 plants from each tray were randomly selected to measure various phenotypic parameters including injury scores, shoot height, root length, and fresh weight. The modified standard evaluation system of IRRI [46] was used to assess the rice seedlings’ susceptibility to salt injury, using a scale of 1 to 9, where 1 indicated high tolerance and 9 indicated high susceptibility (Table 5). Plant survivability was determined as the percentage of surviving plants. The shoot height and root length of rice plants were examined by the conventional measuring method using a ruler. Data were recorded in centimeters. Meanwhile, fresh and dry weights were evaluated and recorded in milligrams. Whole fresh samples were dried in an oven for 3 days at 40 °C before examining dry weight.

### 4.3. Determination of Chlorophyll Contents 

The chlorophyll contents (a, b, and total chlorophylls) were measured using the method described by Quy et al. [47]. Specifically, 100 mg of fresh leaves were placed into a 2 mL microtube, ground with tungsten beads in 1.5 mL acetone (80%) using a tissuelyser and centrifuged at 15,000 rpm for 5 min at 25 °C. Next, 180 µL of supernatant was pipetted into a 96-well plate and the absorbance was recorded at 663 nm (A_663_) and 645 nm (A_645_) using a microplate reader. The chlorophyll a, chlorophyll b, and total chlorophylls were calculated using the following formulas and were expressed in μg/g fresh weight.
Chlorophyll a (µg/g) = 12.7 × A_663_ − 2.69 × A_645_(1)
Chlorophyll b (µg/g) = 22.9 × A_645_ − 4.68 × A_663_(2)
Total chlorophylls (µg/g) = 20.2 × A_645_ + 8.02 × A_663_(3)

### 4.4. Extraction of Plant Samples

Dried plant samples were ground into powder and extracted with 90% methanol for seven days. After filtration, the obtained extract was evaporated at 50 °C using a rotary vacuum evaporator (Rotavapor^®®^ R-300, Nihon Buchi K.K., Tokyo, Japan). The dried methanolic extract was diluted in 100% methanol and then fractionated with hexane at a ratio of 1:1 (*v*/*v*) by a separatory funnel three times. The hexane phase was discarded and the methanolic phase was evaporated at 50 °C under a vacuum to obtain the crude extract. The extracts were then preserved in a fridge at 4 °C for further analysis.

### 4.5. Determination of Total Phenolic (TPC) and Total Flavonoid (TFC) Contents

The total phenolic contents (TPC) of the rice samples were quantified following the Folin–Ciocalteu method [48]. Briefly, a mixture of 20 µL of the sample (in methanol), 100 µL of 10% Folin–Ciocalteu solution, and 80 µL of 7.5% Na_2_CO_3_ (*w*/*v*) were incubated at 25 °C for 30 min. The absorbance was recorded at 765 nm using a spectrophotometer (Multiskan^TM^ Microplate Spectrophotometer; Thermo Fisher Scientific, Osaka, Japan). The TPC was expressed as milligrams of gallic acid equivalent per gram of dry sample weight (mg GAE/g DW).

The total flavonoid contents (TFC) were determined by the aluminum chloride colorimetric method [48]. Briefly, 50 µL of the sample (in methanol) and 50 µL of 2% AlCl_3_ were incubated in the dark for 15 min at 25 °C. The absorbance was recorded at 430 nm and the TFC was expressed as milligrams of rutin equivalent per gram of dry sample weight (mg RE/g DW).

### 4.6. Antioxidant Activity

The antioxidant activity of the rice samples was assessed according to the standard method [48]. Firstly, 80 µL of the sample (in methanol) was mixed with 40 µL of DPPH solution (0.2 mg/mL MeOH) and 80 µL of 0.1 M acetate buffer (pH 5.5). The mixture was then incubated for 20 min at 25 °C in darkness. The absorbance was recorded at 517 nm using a microplate reader. The free radical scavenging activity was calculated as the percentage of reduced absorbance over the control by the following formula:DPPH radical-scavenging activity (%) = (A_c_ − (A_s_ − A_b_)/A_c_) × 100(4)
where A_c_ is the absorbance of the control (MeOH), A_s_ is the absorbance of the sample, and A_b_ is the absorbance of the blank sample (without DPPH).

A linear curve was established by testing various concentrations of the sample and their free radical removal capacities. Based on the obtained equation, IC_50_ values were evaluated, which represents the concentration of the sample required to inhibit 50% of DPPH radicals. The lower IC_50_ value serves as the stronger antioxidant activity.

### 4.7. Identification and Quantification of Phenolics by High-Performance Liquid Chromatography

Eight phenolic acids, gallic acid, protocatechuic acid, vanillin, cinnamic acid, caffeic acid, ferulic acid, *ρ*-hydroxybenzoic acid, and salicylic acid were used as standards. The phenolic profile of the rice samples was identified and quantified using high-performance liquid chromatography (HPLC). The system included a PU-4180 RHPLC pump, LC-Net II/ADC controller, and a UV-4075 UV/Vis (Jasco, Tokyo, Japan) detector. The stationary phase was XBridge BEH Shield RP18 (Waters Corporation, Milford, MA, USA) while the mobile phase included solution A (0.5% aqueous acetic acid) and solution B (100% acetonitrile). The program was operated as 5% B from 0–2 min, 5 to 70% B from 2–12 min, 100% B from 12–16 min and maintained for 6 min, 100% B to 5% from 22–24 min; equilibration was for 10 min. A 5 µL aliquot of the sample was injected into the HPLC system, which was then operated with a flow rate of 400 µL/min for 35 min at room temperature. Phenolic compounds were identified and quantified based on the corresponding peaks and their areas on the HPLC chromatogram compared to those of standard chemicals.

### 4.8. Identification and Quantification of Momilactones by Ultra-Performance Liquid Chromatography-Electrospray Ionization-Mass Spectrometry

The identification and quantification of momilactones A (MA) and B (MB) were performed by ultra-performance liquid chromatography-electrospray ionization-mass spectrometry (UPLC-ESI-MS) [37]. Specifically, the UPLC-ESI-MS system was equipped with an LTQ Orbitrap XL mass spectrometer (Thermo Fisher Scientific, Waltham, MA, USA) and an electrospray ionization (ESI) source. A total of 3.0 μL of the sample (in methanol) was injected by an autosampler (Vanquish autosampler) into the Acquity UPLC^®®^ BEH C18 (1.7 μm, 50 × 2.1 mm i.d.) column (Waters Cooperation, Milford, MA, USA) at a temperature of 25 °C. The flow rate was maintained at 300 μL/min. The gradient of the mobile phase was carried out as follows: 50% solvent A (0.1% trifluoroacetic acid in water) and 50% solvent B (0.1% trifluoroacetic acid in acetonitrile) from 0–5 min, then increased to 100% B from 5–10 min which was retained for 0.1 min, finally the column was equilibrated under the initial conditions for 5 min. The whole process required 15.1 min. The ESI condition was maintained as the ion spray voltage was 4.5 kV, the sheath gas flow rate was 60 and the aux gas flow rate was 20. MS analysis was conducted with a positive Fourier transform mass spectrometer (FTMS) mode with 60,000-resolution as well as 100–1000 m/z of scan range.

Calibration curves for MA (y = 6849.7x − 662,520, r^2^ = 0.9947) and MB (y = 4388.4x − 393,200, r^2^ = 0.9985) using different concentrations of standards MA and MB (100 ng/mL to 5000 ng/mL) were established for quantification of the levels of MA and MB in the tested samples. From the extracted ion chromatogram (EIC), the peak area of samples that matched the retention time of the standard MA or MB was used to calculate the amount of MA and MB in the tested samples (Appendix A Appendix A). The detection (LOD) and quantification (LOQ) limits of MA were 0.29 and 0.89 ng/mL, respectively, and for MB, they were 0.15 and 0.47 ng/mL, respectively.

### 4.9. Statistical Analysis

The experiment was conducted with three replications. Statistical analysis of the raw data was performed through one-way ANOVA and a general linear model using Minitab 16.2.3 software (Minitab Inc., State College, PA, USA). Results were expressed as mean ± standard deviation (SD). Significant differences were determined using Tukey’s test at a 95% confidence level, based on the groups.

## 5. Conclusions

This study is the first to examine the effects of fulvic acid (FA) on improving the salt tolerance of rice seedlings by enhancing their growth parameters, antioxidant activity, and levels of phytocompounds. The results indicate that the most effective concentration for protecting rice seedlings from extreme salt stress of 10 dS/m is a dose of 0.25 mL/L FA (T3 treatment). Among the three rice varieties tested, Nipponbare seedlings showed the strongest tolerance to salt stress in the T3 treatment, followed by Akitakomachi and Koshihikari seedlings. In addition, FA application might increase the accumulation of known salt stress-resistant phenolics in rice, especially salicylic acid. Similarly, a significant reduction in momilactone levels was recorded in rice seedlings under salt stress. However, FA might promote momilactone production in salt-affected seedlings, which was strongly associated with strengthened tolerance of rice to salt stress in this research. Notably, the variation in momilactone and salicylic acid contents over treatments is the same in this study, which suggests momilactones may play an important role in the rice defense system against salinity, similar to salicylic acid. However, the actual contribution of momilactones to the rice response mechanism to salt stress is unclear and needs further comprehensive investigation. The present results imply that FA, an environmentally safe product, is a promising candidate for application in sustainable rice production. At the same time, the effects of FA on different rice varieties and growing stages under a wide range of salinity levels have not been thoroughly indicated and should be elaborated in future studies. Moreover, the application of FA to salt-affected rice fields is also required to confirm its practical implications.

## Figures and Tables

**Figure 1 plants-12-02359-f001:**
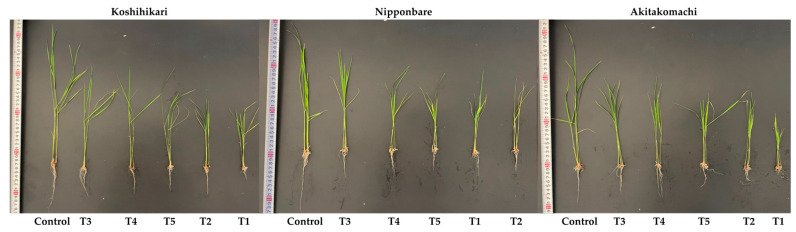
Effect of fulvic acid (FA) treatments on phenotypic performances of rice varieties under 10 dS/m salinity. T1, salinity treatment alone (NaCl 10 dS/m); T2, salinity + 0.125 mL/L FA; T3, salinity + 0.25 mL/L FA; T4, salinity + 0.5 mL/L FA; T5, salinity + 1.0 mL/L FA.

**Figure 2 plants-12-02359-f002:**
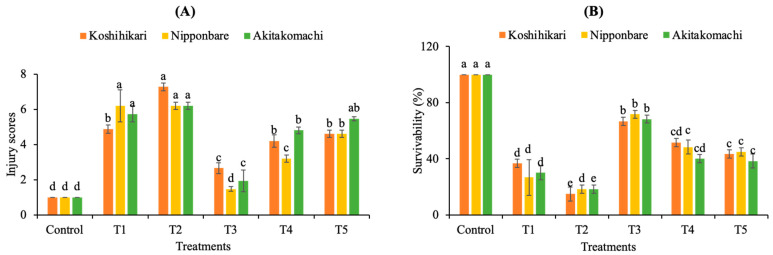
Effects of fulvic acid (FA) treatments on injury scores (**A**) and plant survivability (**B**) at 10 dS/m salinity. T1, salinity treatment alone (NaCl 10 dS/m); T2, salinity + 0.125 mL/L FA; T3, salinity + 0.25 mL/L FA; T4, salinity + 0.5 mL/L FA; T5, salinity + 1.0 mL/L FA. Different letters (a, b, c, d, e) enclosed with columns (same colors) express significant differences at *p* < 0.05.

**Figure 3 plants-12-02359-f003:**
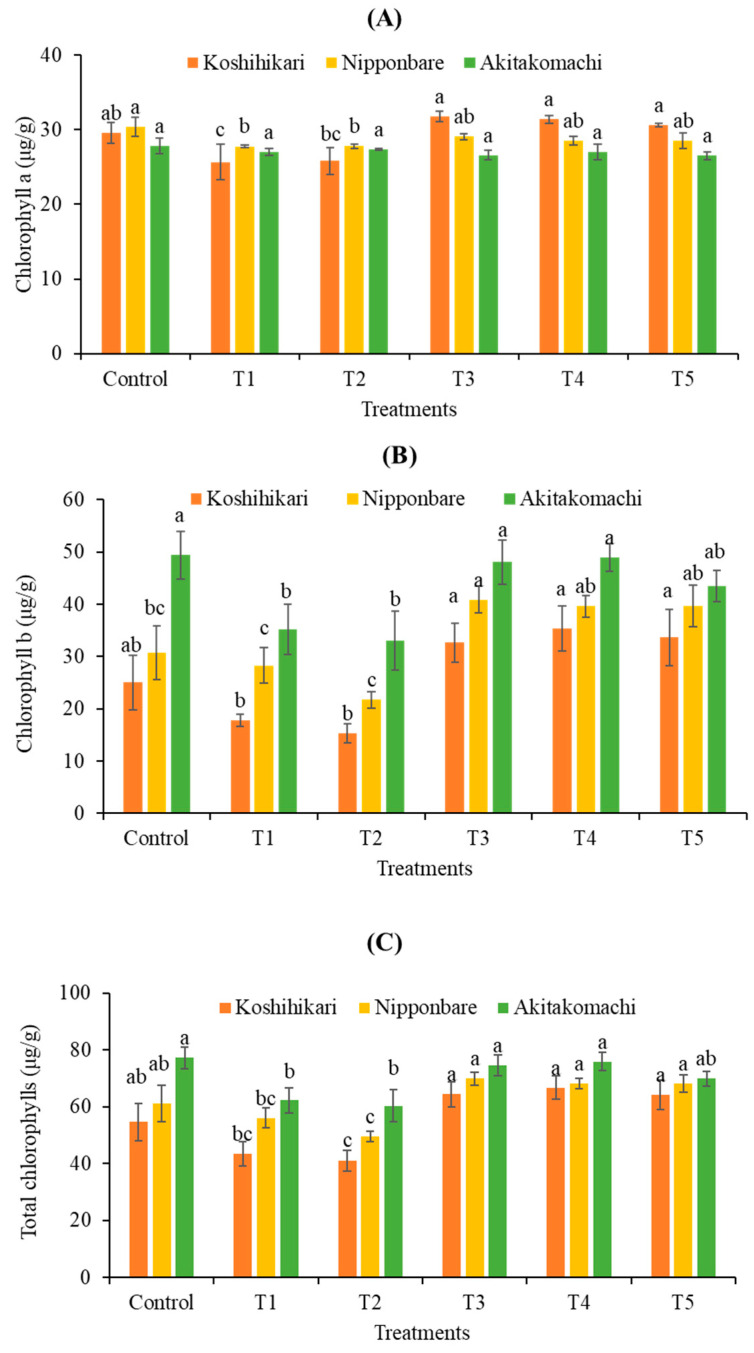
Effect of fulvic acid (FA) on chlorophyll a (**A**), chlorophyll b (**B**), and total chlorophylls (**C**) of rice seedlings. T1, salinity treatment alone (NaCl 10 dS/m); T2, salinity + 0.125 mL/L FA; T3, salinity + 0.25 mL/L FA; T4, salinity + 0.5 mL/L FA; T5, salinity + 1.0 mL/L FA. Different letters (a, b, c) enclosed with columns (same colors) express significant differences at *p* < 0.05.

**Figure 4 plants-12-02359-f004:**
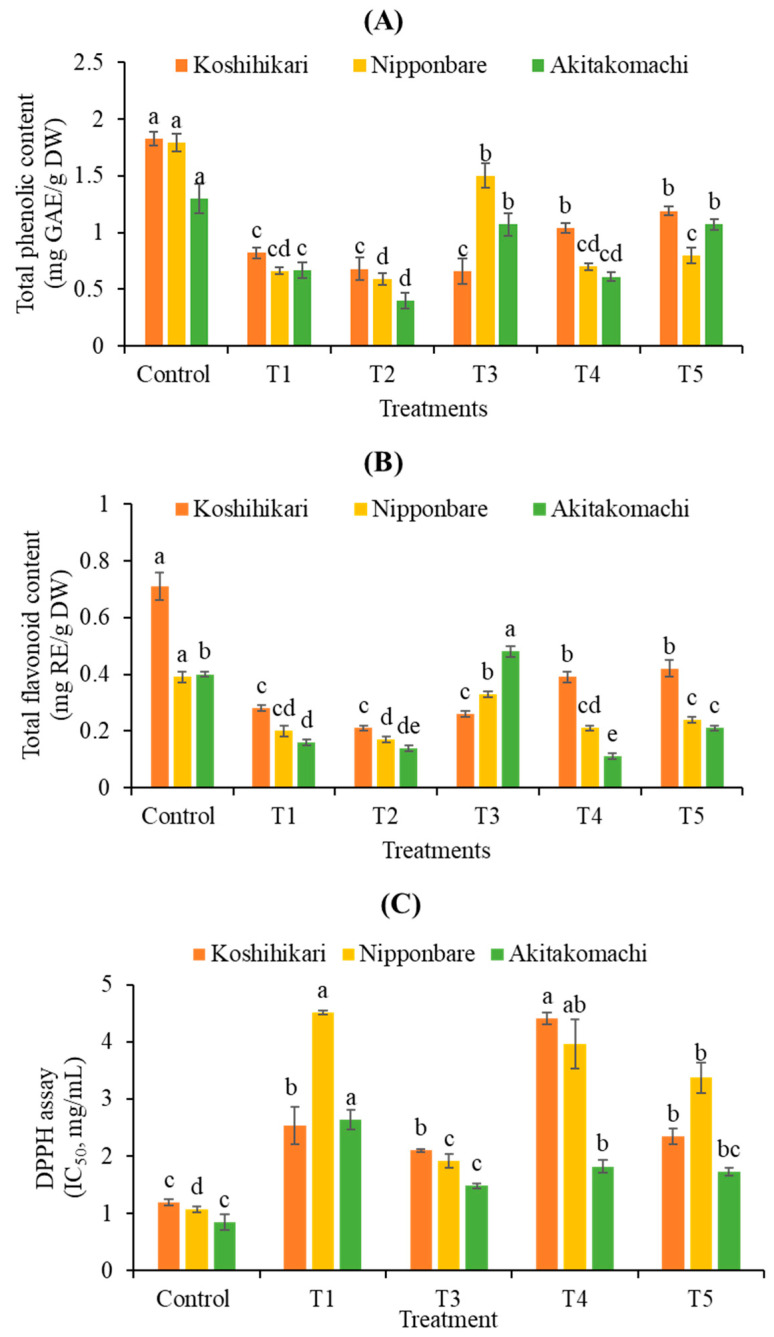
Effect of FA on total phenolics (TPC) (**A**), total flavonoids (TFC) (**B**), and antioxidant activity (DPPH assay) (**C**) of rice seedlings at 10 dS/m salinity. T1, salinity treatment alone (NaCl 10 dS/m); T2, salinity + 0.125 mL/L FA; T3, salinity + 0.25 mL/L FA; T4, salinity + 0.5 mL/L FA; T5, salinity + 1.0 mL/L FA. Different letters (a, b, c, d, e) enclosed with columns (same colors) express significant differences at *p* < 0.05.

**Table 1 plants-12-02359-t001:** Effects of fulvic acid (FA) on phenotypic performances of rice seedlings at 10 dS/m salinity.

Varieties	Treatments	Shoot Height (cm)	Root Length (cm)	Fresh Weight (mg)	Dry Weight (mg)
Koshihikari	Control	26.11 ± 0.47 ^a^	5.92 ± 0.36 ^a^	596.70 ± 47.23 ^a^	96.67 ± 11.55 ^ab^
	T1	4.82 ± 0.60 ^c^	4.69 ± 0.42 ^b^	410.00 ± 10.00 ^cd^	66.67 ± 5.77 ^c^
	T2	12.18 ± 3.29 ^c^	3.27 ± 0.76 ^c^	343.33 ± 66.58 ^d^	6.67 ± 5.77 ^bc^
	T3	23.90 ± 0.20 ^ab^	6.23 ± 0.08 ^a^	540.00 ± 26.46 ^ab^	113.33 ± 11.55 ^a^
	T4	22.05 ± 0.31 ^b^	6.50 ± 0.14 ^a^	470.00 ± 10.00 ^bc^	96.67 ± 11.55 ^ab^
	T5	22.30 ± 1.10 ^ab^	6.19 ± 0.05 ^a^	493.33 ± 20.82 ^bc^	93.33 ± 11.55 ^abc^
Nipponbare	Control	20.39 ± 0.70 ^a^	5.64 ± 0.32 ^ab^	666.67 ± 20.82 ^a^	90.00 ± 10.00 ^ab^
	T1	14.22 ± 0.50 ^c^	3.76 ± 0.32 ^b^	400.00 ± 10.00 ^c^	66.67 ± 5.77 ^b^
	T2	15.59 ± 1.39 ^bc^	6.32 ± 0.96 ^a^	423.33 ± 76.38 ^c^	80.00 ± 10.00 ^ab^
	T3	20.65 ± 0.31 ^a^	5.74 ± 0.47 ^a^	606.67 ± 25.17 ^ab^	106.67 ± 11.55 ^a^
	T4	17.79 ± 1.89 ^ab^	5.27 ± 0.32 ^ab^	420.00 ± 30.00 ^c^	83.33 ± 11.55 ^ab^
	T5	15.62 ± 1.19 ^bc^	4.74 ± 1.22 ^ab^	533.33 ± 20.82 ^b^	90.00 ± 10.00 ^ab^
Akitakomachi	Control	19.65 ± 1.08 ^a^	6.22 ± 0.14 ^ab^	646.67 ± 57.70 ^a^	93.33 ± 5.77 ^ab^
	T1	14.26 ± 0.40 ^c^	4.91 ± 0.26 ^b^	410.00 ± 20.00 ^b^	66.67 ± 5.77 ^c^
	T2	14.75 ± 1.12 ^c^	6.31 ± 0.67 ^ab^	480.00 ± 55.68 ^b^	80.00 ± 10.00 ^bc^
	T3	19.23 ± 0.58 ^ab^	6.87 ± 0.51 ^a^	643.33 ± 20.82 ^a^	103.33 ± 5.77 ^a^
	T4	17.14 ± 0.41 ^b^	4.83 ± 0.43 ^b^	416.67 ± 25.17 ^b^	76.67 ± 5.77 ^bc^
	T5	11.66 ± 0.67 ^d^	2.30 ± 0.91 ^c^	423.33 ± 5.70 ^b^	80.00 ± 10.00 ^bc^
ANOVA					
Variety (V)		***	**	**	***
Treatment (T)		***	***	***	***
V × T		***	*	ns	ns

Data expressed as means ± standard deviation (SD). Different superscript letters within a column in the same variety indicate significant differences at *p* < 0.05 by Tukey’s test. T1, salinity treatment alone (NaCl 10 dS/m); T2, salinity + 0.125 mL/L FA; T3, salinity + 0.25 mL/L FA; T4, salinity + 0.5 mL/L FA; T5, salinity + 1 mL/L FA. *, **, *** denote significant difference at 5%, 1%, and 0.1% probability levels, respectively; ns, not significant.

**Table 2 plants-12-02359-t002:** Effects of fulvic acid (FA) on quantities (mg/g DW) of phenolic compounds in rice seedlings at 10 dS/m salinity stress.

Varieties	Treatments	GA	PCA	VA	SA	CnA	CA
Koshihikari	Control	0.12 ± 0.01 ^a^	0.07 ± 0.00 ^a^	0.23 ± 0.00 ^a^	0.08 ± 0.00 ^a^	0.09 ± 0.01 ^b^	nd
	T1	0.03 ± 0.00 ^b^	nd	0.13 ± 0.00 ^c^	0.04 ± 0.00 ^b^	nd	nd
	T3	0.03 ± 0.00 ^b^	0.04 ± 0.00 ^b^	0.17 ± 0.02 ^b^	0.04 ± 0.01 ^b^	0.16 ± 0.00 ^a^	nd
Nipponbare	Control	0.11 ± 0.01 ^a^	0.05 ± 0.00 ^b^	0.19 ± 0.01 ^a^	0.08 ± 0.00 ^b^	0.08 ± 0.01 ^ab^	nd
	T1	0.03 ± 0.00 ^c^	0.03 ± 0.00 ^c^	0.12 ± 0.00 ^b^	0.03 ± 0.00 ^c^	0.07 ± 0.00 ^b^	nd
	T3	0.06 ± 0.00 ^b^	0.06 ± 0.00 ^a^	0.12 ± 0.02 ^b^	0.25 ± 0.01 ^a^	0.09 ± 0.01 ^a^	0.14 ± 0.01
Akitakomachi	Control	0.10 ± 0.01 ^a^	0.09 ± 0.00 ^a^	0.22 ± 0.01 ^a^	0.10 ± 0.01 ^a^	0.05 ± 0.00 ^a^	nd
	T1	0.04 ± 0.00 ^b^	0.05 ± 0.00 ^b^	0.13 ± 0.02 ^b^	0.02 ± 0.00 ^c^	0.04 ± 0.01 ^a^	nd
	T3	0.04 ± 0.00 ^b^	0.04 ± 0.00 ^c^	0.09 ± 0.02 ^b^	0.05 ± 0.01 ^b^	0.04 ± 0.01 ^a^	0.07 ± 0.01
Variety (V)		***	***	***	***	**	***
Treatment (T)		***	***	ns	***	***	***
V × T		***	***	***	***	***	***

Data expressed as means ± standard deviation (SD). Different superscript letters within a column in the same variety indicate significant differences at *p* < 0.05 by Tukey’s test. T1, salinity treatment alone (NaCl 10 dS/m); T3, salinity + 0.25 mL/L FA. GA, gallic acid; PCA, protocatechuic acid; VA, vanillin; SA, salicylic acid; CnA, cinnamic acid; CA, caffeic acid. **, *** denote significant difference at 1%, and 0.1% probability levels, respectively; ns, not significant; nd, not determined.

**Table 3 plants-12-02359-t003:** Effects of fulvic acid (FA) on momilactone contents (μg/g) in rice seedlings at 10 dS/m salinity.

Treatments	Momilactone A (μg/g)			Momilactone B (μg/g)		
	Koshihikari	Nipponbare	Akitakomachi	Koshihikari	Nipponbare	Akitakomachi
Control	15.47 ± 0.40 ^a^	12.95 ± 0.34 ^a^	7.15 ± 0.38 ^a^	7.42 ± 0.17 ^a^	12.31 ± 0.34 ^a^	7.20 ± 0.10 ^a^
T1	6.93 ± 0.35 ^b^	1.53 ± 0.10 ^c^	1.77 ± 0.08 ^c^	6.37 ± 0.44 ^b^	1.79 ± 0.12 ^c^	1.93 ± 0.07 ^c^
T3	3.06 ± 0.22 ^c^	3.09 ± 0.13 ^b^	5.49 ± 0.18 ^b^	2.48 ± 0.09 ^c^	2.64 ± 0.07 ^b^	3.66 ± 0.20 ^b^
Variety (V)	***			***		
Treatment (T)	***			***		
V × T	***			***		

Data expressed as means ± standard deviation (SD). Different superscript letters within a column indicate significant differences at *p* < 0.05 by Tukey’s test. T1, salinity (NaCl 10 dS/m); T3, salinity + 0.25 mL/L FA; *** denotes significant difference at 0.1% probability level.

**Table 4 plants-12-02359-t004:** Treatment description.

Treatments	Description
Control	Yoshida solution
T1	NaCl (10 dS/m)
T2	NaCl (10 dS/m) and FA 0.125 mL/L
T3	NaCl (10 dS/m) and FA 0.25 mL/L
T4	NaCl (10 dS/m) and FA 0.5 mL/L
T5	NaCl (10 dS/m) and FA 1.0 mL/L

**Table 5 plants-12-02359-t005:** Standard evaluation score (SES) of visual injury.

Score	Observations	Response Category
1	Normal growth, no leaf symptoms	Highly tolerant
3	Nearly normal growth, but leaf tips of a few leaves whitish and rolled	Tolerant
5	Growth severely retarded, most leaves rolled; only a few elongating	Moderately tolerant
7	Complete cessation of growth; most leaves dry; some plants dying	Susceptible
9	Almost all plants dead or dying	Highly susceptible

## Data Availability

Data are contained within this article.

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
