# Peer review of "Fulvic Acid Improves Salinity Tolerance of Rice Seedlings: Evidence from Phenotypic Performance, Relevant Phenolic Acids, and Momilactones"

_plants, 2023, doi:10.3390/plants12122359_

Round 1
Reviewer 1 Report
Study is focused on the potential effect of fulvic acid on salinity tolerance of some crops. Authors showed that fulvic acids enhanced salinity tolerance, stimulated their growth, antioxidant activity, and chlorophyll contents.
Figure 1: please give reference, is it model formula? fulvic acids are a mixture of many constituents...
How were measured shoot height and root length? Were plants scanned? Was used image analysis?
What is the mechanism of improvement of salinity tolerance?
Author Response
Dear Respected Reviewer,
We sincerely thank you for your valuable comments and suggestions. The manuscript has been extensively revised following your advice. Please kindly see the attachment that includes our detailed responses to your comments.
Sincerely thanks,
Tran Dang Xuan
Corresponding author
On behalf of all authors

Reviewer 2 Report
Fulvic Acid Improves Salinity Tolerance of Rice Seedlings: Evidence from Phenotypic Performance, Relevant Phenolic Acids, and Momilactones
The manuscript is written in very poor language. I suggest to give special consideration to this point. Additionally, the novelty of the study is also questionable. The quality of the data, its presentations and write up are very poor. I am unable to accept the MS in the present form.
The following points may help improve the quality of the MS.
The abstract is a bit shallow. Start the abstract with background problem i.e. salinity. More details of the treatment should be given. Increase/ decrease should be given in quantitative values (%).
Introduction
There is no need for Figure 1. Chemical structure of fulvic acid.
Hypothesis and objectives should be well described.
Material and Methods
The description of the treatments is not appealing. It should be Control, T1, T2, T3… Salinity treatment should be T1.
Made and model of pH meter should be mentioned.
A part of the fresh samples was dried in an oven for 3 days at 40 °C and dry weight was measured in milligrams. Why not the whole sample was dried?
Results and Discussion
Among the three varieties, Nipponbare seedlings reveal the best tolerance with the lowest injury score (1.47) and highest survivability (71.67%) under the salinity stress of 10 dS/m (Figure 3). I Don’t agree with this observation as the results are statistically non-significant.
In Fig. 4, chlorophyll a (A), chlorophyll b (B), and total chlorophylls (C) should be mentioned on y-axis of the graphs.
Line 150; what is salted control at 10 dS/m??
Figure 5. The parameters should be well described on y-axis of the graphs. Moreover, T1 is missing in Fig 5C.
Line 277-284 Seems irrelevant to the context of the study. Discussion should be more conclusive supported with the pertinent references.
Conclusion should also be more comprehensive with limitations and future perspective.
Should be improved
Author Response

(The authors gave the same response as above.)

Reviewer 3 Report
This work investigated the influence of Fulvic Acid on salinity tolerance of rice seedlings.The topic is interesting and well within the aims of the Journal.
It is really well written, the materials and methods are described in depth, the results are clear and well represented in graphs and tables and both the discussion and the references are exhaustive of the topic.
Only the design research could be further improved, even in future scientific works, by comparing different levels of salinity in order to understand if the response of the different cultivars remains the same or changes. Furthermore, the reasons that led to the choice of using an EC level of 10dS/m are not deduced from the manuscript, therefore I suggest adding this explanation in the text.
Author Response

(The authors gave the same response as above.)

Round 2
Reviewer 2 Report
The authors have improved the MS and now it is acceptable for publication